# An Evaluation of Temporal Distributions of High, Low, and Zero Cohort Morbidity of Cumulative First Treatment Bovine Respiratory Disease and Their Associations with Demographic, Health, and Performance Outcomes in US Feedlot Cattle

**DOI:** 10.3390/vetsci10020089

**Published:** 2023-01-24

**Authors:** Blaine Johnson, Brad White, Phillip Lancaster, Robert Larson

**Affiliations:** 1Department of Diagnostic Medicine and Pathobiology, College of Veterinary Medicine, Kansas State University, Manhattan, KS 66505, USA; 2Department of Clinical Sciences, College of Veterinary Medicine, Beef Cattle Institute, Kansas State University, Manhattan, KS 66505, USA

**Keywords:** feedlot, bovine respiratory disease (BRD), hierarchical clustering, epidemiology

## Abstract

**Simple Summary:**

Cluster analyses were performed on the cohort temporal distributions of bovine respiratory disease BRD. Results illustrated that the optimal number of clusters differed by the level of morbidity within the cohort. Different temporal patterns of cumulative first treatment BRD which could influence disease prevention and control techniques were identified. Descriptive and statistical associations of risk factors for each cluster better describes the cattle represented within each cluster. More research is needed to understand the potential economic impact each cluster has on feedlot production. Improved understanding of cohort timing and magnitude of BRD could potentially identify interventions to mitigate disease burden and economic impacts.

**Abstract:**

Timing and magnitude of bovine respiratory disease (BRD) can impact intervention and overall economics of cattle on feed. Furthermore, there is a need to better describe when cattle are being treated for BRD. The first objective was to perform a cluster analysis on the temporal distributions of cumulative first treatment BRD from HIGH (≥15% of cattle received treated for BRD) and LOW cohorts (>0 and <15% of cattle received treated for BRD) to assess cohort-level timing (days on feed) of BRD first treatments. The second objective was to determine associations among cluster groups (temporal patterns) and demographic risk factors, health outcomes, and performance. Cluster analysis determined that optimal number of clustering groups for the HIGH morbidity cohort was six clusters and LOW morbidity cohort was seven clusters. Cohorts with zero BRD treatment records were added for statistical comparisons. Total death loss, BRD morbidity, average daily gain (ADG), railing rate, days to 50% BRD, cattle received, shrink, arrival weight, and sex were associated with temporal groups (*p* < 0.05). These data could be used as a tool for earlier identification and potential interventions for cohorts based on the BRD temporal pattern.

## 1. Introduction

Bovine respiratory disease (BRD) has remained the most economically important disease in feedlot cattle for over the past half century [1,2,3,4]. The level of BRD morbidity and mortality has not dramatically decreased, and some report increased rates of mortalities, even with the antimicrobial therapies and management strategies currently available [5,6]. There is growing concern cattle are exhibiting increased number of treatments for BRD later in their respective feeding period [7,8]. Bovine respiratory disease affects cattle typically within the first 45 days of feed [9]. However, many changes from the economic to genetic levels, impact how cattle are fed and the duration of their respective time at a feedlot. Cattle that have late-day feeding BRD carry the potential of an increased risk of death, which incurs larger financial burden (feed, cost of treatment, yardage, and interest) compared to the same animal experiencing early-onset BRD at the feedlot.

The timing of cumulative first BRD treatments has been previously reported and analyzed [10]. However, there are gaps in knowledge of potential differences in BRD temporal patterns based on overall magnitude of BRD within a cohort. A better understanding of the timing and magnitude of BRD within cohorts, along with the association of risk factors, could provide economic advantages for producers through decreased health problems and increased performance. In consideration of these goals, two study objectives were developed. The first objective was to perform two cluster analyses on the distributions of cumulative first treatment BRD, one on the HIGH cohort (≥15% of cattle received treated for BRD) and one on the LOW cohort (<15% of cattle received treated for BRD) based on the timing of BRD treatments (days on feed). The second study objective was to determine potential associations between cluster groups (temporal patterns) and demographic risk factors (arrival characteristics), health outcomes (morbidity, mortality, and railing rate) and performance (ADG).

## 2. Materials and Methods

Animal Care and Use Committee approval was not obtained for this study due to retrospective commercial data being obtained from privatized databases.

Data from 10 commercial feedlots in the central US were used in the current study. Study data encompassed cattle placed on feed from 2 January 2019 to 15 December 2021. Individual treatment and cohort data were downloaded from privatized commercial feedlot software into R^®^ studio [11] (https://www.r-project.org/) for data wrangling, cleaning, and analysis. The current study used a case definition of a cohort defined as a group of cattle purchased within a similar timeframe and managed similarly to a common endpoint but not necessarily fed in the same pen throughout their feeding period. For this study’s purposes, a BRD treatment was defined as cattle identified with clinical BRD and treated with an antimicrobial approach. The association analysis included cohorts with no or zero BRD treatments. However, cohorts with zero BRD treatments were not included in the temporal distribution cluster analysis since groups were clustered according to their timing of first BRD treatment.

### 2.1. Data Management: Variable Creation

A new variable was created using yard identification, lot identification number, pen, and arrival year to have a unique identification number, “uid2”, to tie individual treatment records to the cohort level data. For individual record data, a new variable was created to represent days on feed at event by taking the event date (treatment date) and subtracting the arrival date (days on feed = event date − arrival date). Variables were created for cohort level data that represented total number of BRD treatments within each cohort both as a number and as a percentage of the cattle received. Another variable was created that grouped individual animal treatment data to cumulative daily cohort BRD treatments, expressed as percentage of cattle received, “cumulative BRD percent”. A variable was created to represent the cohorts’ timing to 50% BRD treatments by taking individual BRD treatment records and determining the days on feed at which 50% of cumulative BRD treatments were reached. A variable for arrival quarter “in_qrt” was created by taking the month of arrival and categorizing it into 1 (January through March), 2 (April through June), 3 (July through September), or 4 (October through December). Shrink was calculated by taking average cohort payweight minus the average cohort arrival weight and dividing by payweight then multiplying by 100 to get the percentage, (shrink, % = ((payweight − arrival weight)/payweight) × 100). Performance variables were created from cohort closeout. Only animals that were shipped for harvest were used in the performance variable calculations (deads out). A variable for average daily gain (ADG) was created from the difference between average cohort shipping weight (deads out) and average cohort arrival weight divided by the number of days on feed (DOF) (ADG = (shipping weight − arrival weight)/DOF).

### 2.2. Cluster Analysis

Individual animal treatment records from the defined study period were obtained for each cohort. Both cluster analysis utilized cohorts with total pen morbidity of ≤100% for the first treatment for BRD. The raw data had two cohorts with >100% pen morbidity, which were removed from analysis. Authors used 15% total cohort BRD morbidity as a threshold to label cohorts as HIGH or LOW BRD morbidity. The 15% distinction between HIGH and LOW was previously utilized in other published work [12]. Cohorts with HIGH BRD morbidity had ≥15% total cohort BRD morbidity. Cohorts with LOW BRD had total cohort BRD morbidity between >0% and <15%. Cohort BRD morbidity was calculates by dividing the total number of cattle treated for first treatment BRD by the initial cattle received.

Two cluster analyses were performed, one for HIGH and one for LOW morbidity groups; no cluster analysis was performed on BRD timing for the ZERO morbidity group as there were no treatments. Individual first treatment BRD data were aggregated to daily cohort level cumulative BRD morbidity (percent BRD treatments per cattle received) data for cluster analysis. Cohorts were clustered on temporal distributions (days on feed) of cumulative BRD treatments, expressed as a percent of cattle received (cumulative BRD treatments/initial cattle received) × 100) using the hclust function from the stats package in R [11]. Cluster distancing was performed utilizing the “ward.D” method [13]. Clustering groups were evaluated similarly to [10]. In short, data were clustered across multiple sequential numbers of clusters and computed cluster heights were plotted by creating an elbow plot in ggplot2 [14]. These graphical figures illustrate the error of the sum of squares (*y*-axis) against the successive number of clusters (*x*-axis) to determine the step (cluster) at which further reduction does not result in a substantial change in height. Optimal cluster groups were determined when relative change in height values were first minimized.

### 2.3. Descriptive Statistics and Association Analysis

Descriptive statistics were computed for each clustering group within HIGH and LOW and the group containing ZERO BRD treatments (“0”). For each cluster, a mean, standard deviation and range were computed using cohort performance and individual health data. Medians were calculated for cohort BRD morbidity and timing to 50% BRD treatment days on feed. Frequency tables were created for non-continuous data.

Average cohort arrival weight was categorized into seven 45.5 kg categories (“181 kg to 226 kg”, “227 kg to 272 kg”, “272.1 kg to 317 kg”, “318 kg to 362 kg”, “363 kg to 408 kg”, “408 kg to 453 kg”, “454 kg to 498 kg”) to avoid lack of model convergence and for external generalization. Weight groups separated by 45.5 kg are commonly used to describe cohorts of cattle within the feeding industry. Clustering groups were concatenated into one factor with 13 levels and ZERO (H1, H2, H3, H4, H5, H6, L1, L2, L3, L4, L5, L6, L7 and 0) for the statistical analysis. The number of clustering groups were determined by the cluster analysis, with each level exhibiting similar timing of BRD treatment and the magnitude of BRD represented by “H” for HIGH, “L” for LOW, or “0” for ZERO as previously described. This allowed for a statistical comparison between all cohort groupings.

Association analyses were performed in R programming using the lmer4 package [11,15]. Linear and logistic regression mixed models were created to compare the association of arrival characteristics and performance characteristics to clustering groups. Descriptive statistics were created to determine association of shrink and cohort size (number of cattle received) at arrival with clustering group. Multivariate models were created to determine the association of clustering groups with performance (ADG) and health (total death loss percentage, BRD morbidity, total head railed (cattle removed from a cohort prior to the cohort’s shipment date), and days to 50% BRD treatments) factors. Covariates used in each multivariate model included arrival weight class category, arrival quarter, and sex. The BRD morbidity model also included ADG as a covariate. All continuous outcomes were assessed for linearity. Any outcomes identified as non-linear were transformed by taking the log or square root of the outcome and assessing it for overall fit for the final model. The best fit model from transformed data is reported in the results. All models were built using the backwards eliminations technique [16]. Any of the above covariates that were not significant (*p* > 0.05) were removed from the final model. Multivariate models were assessed for collinearity utilizing the “performance” package “check_collinearity” function in R^®^ [17]. No variables were identified as being colinear.

Feedlot was used as random intercepts to account for hierarchical data structure and lack of independence of lot within feedlot for both descriptive and multivariate analysis. Model comparisons were created using the emmeans package [18] using a Tukey Kramer adjustment for multiple comparisons. Model predictions were weighted by cells using method = “cells” in emmeans. Models which required outcomes to be transformed to meet the assumption of linearity were back-transformed within the model estimates in the post hoc comparisons using emmeans. A significance of *p* < 0.05 was determined a priori.

## 3. Results

The final dataset used to model demographic and risk associations and cluster analyses contained 7735 cohorts representing 1,016,873 fed cattle. There were 5903 (76.3%; 5903/7735) cohorts in the low morbidity group, 1597 (20.6%; 1597/7735) cohorts in the high morbidity group, and 235 (3.01%; 235/7735) cohorts which reported no morbidity due to BRD. There were 66.1% heifers (5113/7735) and 33.9% steers (2622/7735). Average cohort arrival weight was 343 kg and ranges from 182 kg to 497 kg. The number of cattle received per cohort averaged 132 cattle and ranged from 40 to 892 cattle per cohort. Average cohort overall death loss was 1.3% and ranged from 0% to 43.1% across the enrolled cohorts. The average feeding length (DOF) per cohort was 157 days and ranged from 60 days to 350 days. On average, cohorts had a 2.1% shrink; however, shrink ranged from −94.9 to 25.1%. There were 950 cohorts with negative shrink, with two cohorts reporting a shrink of less than −10%. A negative shrink percentage indicates an average cohort weight gain from purchase to arrival at feedlot in the current study. Average time to reach 50% cohort BRD morbidity was 32 days and ranged from 0 to 197 days on feed.

### 3.1. Cluster Analysis

Results from the cluster analysis determined that the optimal number of clustering groups for the HIGH morbidity cohort was six clusters and LOW morbidity cohort was seven clusters. For the LOW clustering groups, increasing from seven to eight indicated little distance between clusters, and reducing from seven to six clustering groups resulted in dramatic curve changes, which is suggestive of a combination of two dissimilar clusters [10,13]. Figure 1 and Figure 2 show the distance (*y*-axis) vs. the number of cluster groupings (*x*-axis).

The distance between clustering groups indicates dissimilarity; the greater the number of groups that are clustered together, the smaller the distance between groups, indicating minor differences between groups. For the rest of this study, HIGH was divided into H1 through H6 and LOW into L1 through L7 to represent their respective clustering groups.

The HIGH cluster groups resulted in varied patterns of BRD morbidity timing, with all groups reaching 50% of morbidity by 60 DOF (Figure 3). Disease onset was rapid in many of the HIGH clusters, with 75% of morbidity achieved by day 50 in H1, H2, H3, and H5. The cluster H6 displayed a more delayed pattern, with ~20% of morbidity occurring after 100 DOF.

The LOW cluster groups resulted in more timing variability relative to BRD disease patterns compared to the HIGH groups (Figure 4). Three of the clusters (L5, L6, L7) did not reach 50% of morbidity until after day 60. The LOW group clusters had a range in achieving 50% morbidity from day 13 to day 109 after arrival.

Cohorts were placed in clusters based on morbidity timing (or in the case of ZERO, the lack of BRD morbidity). Table 1 shows the descriptive statistics of cohorts within clustering groups and by factors of interest.

### 3.2. Risk Association Analysis

The association of cohort arrival demographic risk factors with the clustering group is reported in Table 2. The number of cattle received was also found to be associated with clustering groups (*p* < 0.01). No differences were found within HIGH clusters in the estimated cattle received. Cohorts in ZERO had the lowest estimated cattle received per cohort at 93 (SE = 8.22). Cohorts in L2 had an estimated cattle received per cohort of 145 (SE = 7.31), which was greater than all of HIGH, ZERO, L1, and L7, but similar to L3 to L6.

Shrink was found to be associated with clustering group (*p* < 0.01). Clusters in LOW (L1 through L7) had a smaller estimated arrival shrink compared to cohorts in HIGH (H1 through H5). Cohorts in ZERO cluster had the lowest shrink loss at 1.24% (SE = 0.39). However, ZERO were no different than cattle in H6, L5, and L7. Shrink was greatest for cohorts placed in H1 (3.65, SE = 0.32) but different compared to H6 (2.12, SE = 0.31) within the HIGH group, as well as different compared to LOW and ZERO.

Arrival weight was found to be associated with clustering groups (*p* < 0.01). Cohorts in HIGH had lower arrival weights compared to cohorts in LOW or ZERO. Cohorts in H4 and H6 reported the lowest arrival weight at 295 kg and 295 kg, respectively (SE = 8.3 and 8.2). Cohorts in H6 were no different compared to H3 and H1. Cohorts in ZERO reported the heaviest arrival weights at 376 kg (SE = 9.1), which was significantly greater than all cohorts except L7.

Finally, the probability of a cohort being steer was associated with clustering group (*p* < 0.01). Cluster L6 had a lower probability 0.27 (SE = 0.03) of being steer compared to H1, H2, L1, and ZERO (0.40 and 0.35, 0.35 SE = 0.04, 0.04, 0.3 and 0.04, respectively).

Table 3 displays the association analysis of performance and health metrics with clustering groups. Average daily gain (ADG) was associated with clustering group (*p* < 0.01). Cohorts in clusters H1 through H6 had lower reported ADG compared to L2 through L7 and ZERO. Cohorts in ZERO had the highest ADG at 1.66 kg/d (SE = 0.05). Cohorts in H6 had the lowest reported ADG at 1.33 kg/d (SE = 0.05); however, it was not different compared to other cohorts within H1 through H5.

Bovine respiratory disease was associated with clustering group (*p* < 0.01). Cohorts in HIGH (H1 through H6) reported higher estimated BRD morbidity compared to cohorts in LOW (L1 through L7) and ZERO by design. H1 was estimated to have the highest predicted BRD morbidity at 27.25% (SE = 0.82). ZERO has the lowest estimated BRD (0.01, SE = 0.02), which was found not to be different compared to L7 at 2.33% (SE = 0.32).

Total death loss was associated with clustering group (*p* < 0.01), with higher death loss estimated for cohorts in H1 through H5 vs. LOW (L1 through L7), H6, and ZERO clustering groups. H1 had the greatest model estimated death loss at 4.19% (SE = 0.26) but was not different compared to H2 (4.15%, SE = 0.25). Cohorts in cluster group ZERO (1.15%) were not different from L7 (1.84%) but had lower death loss than all other groups. There were no differences in death loss between the cohorts in cluster groups H6 and in all of the LOW cluster groups. Cohorts in ZERO reported the lowest total death loss at 0.93% (SE = 0.27); however, this was not different compared to L7 (1.64%, SE = 0.32).

Clustering group was associated with number of railed cattle (*p* < 0.01). Model-estimated rail rates were greater for HIGH (H1 through H6) clusters compared to LOW (L1 through L7) clusters and ZERO. Cluster H2 had the highest estimated railing rate of 13.6 rails per 1000 heads received (SE = 3.7); however, this was not different compared to H1 and H3 at 11.9 and 11.2 per 1000 cattle received (SE = 3.3 and 2.7; *p* > 0.05), but greater than all other cohorts. Cohorts in the ZERO clustering group had the lowest number of estimated rails per 1000 cattle received at 1.8 (SE = 0.5; *p* < 0.05), which was not different from the L7 clustering group at 2.8 cattle per 1000 cattle received (SE = 1.0). Estimated railing rates were lower in L1 and L6 compared to L2 (4.0 and 3.6 cattle per 1000 cattle received (SE = 1.1 and 1, respectively) vs. 5.1 cattle per 1000 cattle received, SE = 1.4, respectively).

Days to 50% cohort morbidity was found to be associated with cluster group (*p* < 0.01). There were differences between HIGH and LOW clusters and within HIGH and LOW clusters. Cohorts in H6, L5, L6, and L7 had estimated days to 50% cohort BRD at 59.8 (SE = 1.77), 61.2 (SE = 0.82), 60.6 (SE = 0.63) and 109.8 DOF (SE = 2.19), respectively. These clusters were much later compared to H1 through H5 and L1 through L4.

## 4. Discussion

These data identified BRD temporal patters at two levels of cohort morbidity (HIGH ≥ 15% and LOW > 0 to <15%). In addition, these data showed several important risk factors associated with clustering group, such as shrink, number of cattle received, and sex. Performance and health metrics were also found to be associated with clustering group when adjusting for other factors, such as ADG, BRD morbidity, death loss, and rail cattle. The creation of these temporal distributions and model estimates may lead to future research to predict when cohorts would experience BRD. Hierarchical clustering methods have been previously employed to describe temporal patterns of BRD in feedlot cattle [10]. One of the weaknesses of the previously described temporal patterns was the inability to relate the magnitude of BRD of a cohort within its temporal distribution pattern. Babcock et al. (2010) reported that assessing both the timing and magnitude of the disease leads to clusters being driven exclusively on magnitude of disease [10]. Authors of the current study also experienced the previously stated conundrum as a major obstacle. Efforts to address this obstacle led authors to cluster feedlot cohorts into HIGH or LOW BRD to better stratify the magnitude of BRD within temporal distributions. The current study set a threshold of 15% to distinguish cohorts that were categorized as HIGH or LOW, similarly to previous research [12]. Another novelty of the current research was incorporating cohorts reporting no BRD morbidity and not limiting the timing of BRD for inclusion of cohorts. This allowed for comparisons of cohorts to be grouped by the timing of BRD or lack thereof and determine if there were differences between cohort demographics and performance of cattle within the cohorts fitting each group based on the timing of magnitude of BRD.

Babcock et al. (2010) reported seven clusters that optimally fit their study data, and the current study found seven clusters were optimal in LOW, with six optimal in HIGH [10]. The major difference between the two studies was accounting for BRD magnitude. The current study partitioned BRD morbidity into two groups, which resulted in 13 distinct groupings for cohorts based on the timing of BRD and one group of ZERO BRD morbidity.

For arrival demographic associations with clustering group, the current study analyzed cohort size (number of cattle received), shrink, arrival weight, and sex in univariate analyses with cohort clusters. LOW (L1 through L7) clusters were estimated to have greater cohort sizes with greater average number of cattle received; however, this was not the case for ZERO cohort sizes. Model-estimated average cohort size for cattle in ZERO grouping was 93 cattle, which resulted in the lowest estimated average cohort size overall, though not different from L7 or H6. This could be due to groups of local, ranch direct cattle kept in a similar cohort, since ZERO also reported a low estimated shrink (1.24%, SE = 0.39). The ZERO cohort may have comprised cohorts backgrounded or managed differently prior to entering the feedlot, and older, due to a higher arrival weight of 376 kg (SE = 9.1 kg).

Shrink was lowest in cohorts reporting no BRD morbidities (ZERO) and highest in HIGH (H1 through H5) cluster groups. Findings showing higher levels of shrink associated with higher morbidity were similar to previous research, which reported that cattle with greater shrink (body weight loss from purchase to feedlot arrival) had greater BRD morbidity risk as well as lower arrival body weights [19,20]. The authors suggested this was likely due to the stress incurred during transport and distance traveled.

Arrival weights were estimated to be heavier for the LOW cohort in L1 through L7 clusters or ZERO compared to cohorts in HIGH clusters, and cohorts with ZERO morbidity had the largest estimated average arrival weight compared to all other clusters except L7. Clusters in H1 through H6 had the lowest estimated average cohort arrival weight compared to LOW and zero clusters. In a literature review on the epidemiology of BRD, Taylor et al. (2010) [19] reported several studies that associated lower body weights with higher BRD morbidity, which was also consistent with other previous literature [10,19]. Thus, the HIGH clusters have overall lower weights and carry higher risk for BRD morbidity.

The timing to 50% BRD was different between the two clusters (H1 = 10.1 days on feed to 50% cohort BRD vs. H4 = 40 days on feed to 50% cohort BRD). Differences in BRD magnitude and timing between the H1 and H4 groups could be due to different pathogens or backgrounding methods (vaccination status, nutritional status, etc.), none of which could be controlled for in the current study.

Model estimates for ADG reported that LOW L2, L3, L4, L6, and ZERO outperformed all the HIGH clusters (H1 through H6). This is consistent with other research reporting that ADG was negatively associated with BRD; the hypothesized cause was presumably due to anorexia and inflammation [21].

Estimates for average cluster morbidity showed variation within the HIGH and LOW clustering groups. Data were split by cohort morbidity; thus, differences between HIGH and LOW were to be expected. However, the variation within each HIGH and LOW clustering groups is interesting because it could suggest that the timing of BRD treatments could potentially be related to the amount of BRD a cohort experiences. HIGH groups that were clustered in H4, H5, H6 experienced >50% of their BRD after 30 days compared to H1, H2, and H3 which experienced 50% of their BRD by 22 days on feed. Potential reasons for the reported differences in timing to 50% disease could be a pathogen causing diseases such as viral vs. bacterial. However, literature on the timing of clinical disease from observational research appears to be limited. Experimentally induced disease typically appears clinically sooner than 21 days for both viral and bacterial agents [22,23,24].

Differences in cohort level morbidity could be associated with the amount of stress from transport if shrink is considered a measurable metric of stress due to transport. Clusters H1 through H3 had higher numerically estimated shrink compared to H4 through H6. Total death loss was found to follow similar patterns as BRD. The amount of death loss attributed to BRD has previously been reported to range from 41 to 52% of the dead population [25]. Since morbidity of BRD follows similar death loss patterns, it is likely that most death loss is driven by BRD. Total death loss was lower for cohorts in LOW (L1 though L7) and ZERO compared to HIGH (H1 through H5), except for H6 which was not different than LOW but greater than ZERO. Death loss was not different in the LOW groups; however, ZERO BRD morbidity demonstrated the lowest death loss, which was different from L1 through L6 but not different from L7. Thus, it would appear that with greater shrink came a greater amount of BRD at earlier days on feed.

Railing rate estimates appear to be higher in HIGH (H1 through H6) vs. LOW (L1 through L7) and HIGH (H1 through H6) vs. ZERO clusters. The rates for railing cattle do not follow the same patterns as BRD morbidity or death loss. Thus, cattle are being railed likely for other diseases outside BRD, such as lameness/muscular-skeletal, which was reported in previous research [26]. Cattle in low clustering groups were estimated to have fewer cattle railed (per 1000 cattle received) compared to HIGH cluster groups. These data reported that rates differed more in cohorts that were within the HIGH clusters, with cluster H2 having the highest rail rate. In addition, differences in railing rates were reported in estimates within the LOW cluster groups. Arrival weights with HIGH had a lower estimated arrival weight vs. LOW and ZERO. ZERO had the heaviest average cohort arrival weight of 376 kg (SE = 9.1).

Model-estimated DOF to 50% cumulative BRD first treatment drastically differed between HIGH and LOW clusters and notably within the respective clusters. Four cluster groups stand out in this analysis: H6, L5, L6, and L7, which were estimated to hit 50% of their first BRD treatment by day 59.8, 61.2, 60.6 and 109.8, respectively on feed. These clusters would be considered much later than typical for the industry. The authors of [27] reported that 74% of BRD cases occurred within the first 42 days on feed, which many of the cluster groups achieved [27]. However, a few clustering groups were outside 42 days on feed, which could bear relation with what the industry is describing as mid- to late-day BRD [8,28]. The LOW clusters had overall lower morbidity, which means that even regardless of timing, fewer cattle were treated. Thus, when comparing the percentages of cattle treated, one must consider the total number of cattle treated (denominator) relative to the timing of treatments.

In consideration of the current study, these data are retrospective and observational, meaning they carry a greater inherent risk of bias [29,30,31]. Caution should be used when generalizing these data to other populations not like the current study’s population. Furthermore, this study’s population contained a higher-than-average heifer population compared to most feedlot operations. Operational data are known to be susceptible to confounding [32]. Measures were taken to remove incomplete or missing data. Risk factor inclusions were limited to available data. Other important risk factors such as previous location/distance travel, genetics, number of intact males in steer cohorts, weather, and carcass performance were not included in the current analysis as with previous research [19].

Furthermore, the objective was to describe when BRD occurred during the feeding period while incorporating the magnitude of cohort morbidity from the study’s population. Previous work utilized a technique to limit the evaluation period to 100 days after arrival, resulting in all cohorts with cattle being at risk for each day in the study [10]. A limitation of the current work is the evaluation period (up to 275 days), which included days when cohorts were not at risk for additional BRD treatments as they had been sold. The study’s objective was to determine temporal risk patterns during the entire feeding period; thus, an evaluation of the entire period was selected. After cattle left the facility, their number of BRD treatments did not change (increase or decrease). Thus, these data were modeled to strictly evaluate the days on feed of when a cohort’s first treatment BRD occurred.

## 5. Conclusions

By splitting the data into HIGH and LOW morbidity groups, we were able to show different temporal patterns of cumulative first treatment BRD while giving some reference to the magnitude of BRD within the cohorts represented in their respective cluster. The descriptive and statistical associations of risk factors for each cluster better describes the cattle represented within each cluster. These data have potentially identified and described clusters that could fit the industry’s concern for late-day BRD. Three of the clusters are in the LOW morbidity group, with one being in the HIGH morbidity group (L5, L6, L7 and H6). Cohorts never treated for BRD had the lowest shrink, lowest number of railed cattle, heaviest average cohort arrival weight, and reported no BRD treatments in the raw data. More research is needed to understand the potential economic impact each cluster has on feedlot production. In addition, these data could be used in predictive modeling; if one could predict a cohorts timing and magnitude of BRD, one could potentially identify interventions to mitigate disease burden and economic impacts.

## Figures and Tables

**Figure 1 vetsci-10-00089-f001:**
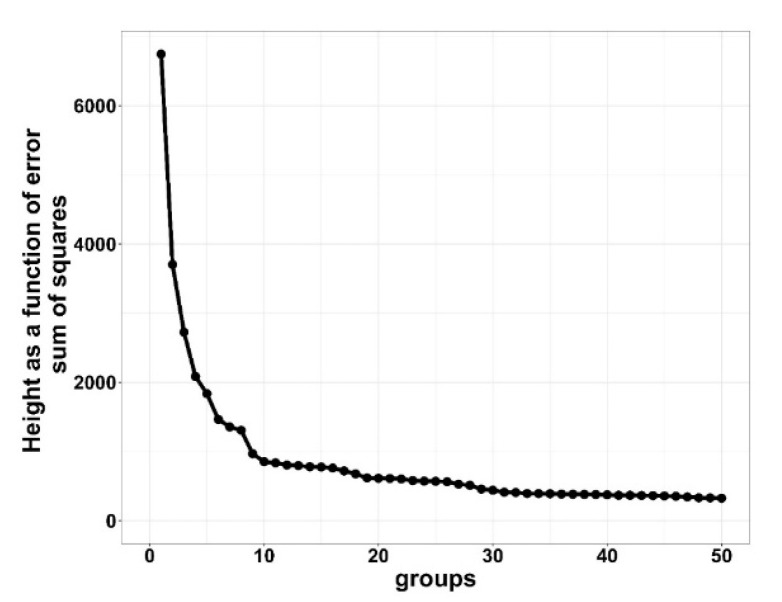
Elbow plot of HIGH (≥15%) bovine respiratory disease (BRD) morbidity cohorts demonstrating height as a function of error sum of squares for clustering cohorts with bovine respiratory disease from 10 US commercial feedlots. Distance between points is a difference in height of error sum of squares. Greater heights are suggestive of larger differences between groups.

**Figure 2 vetsci-10-00089-f002:**
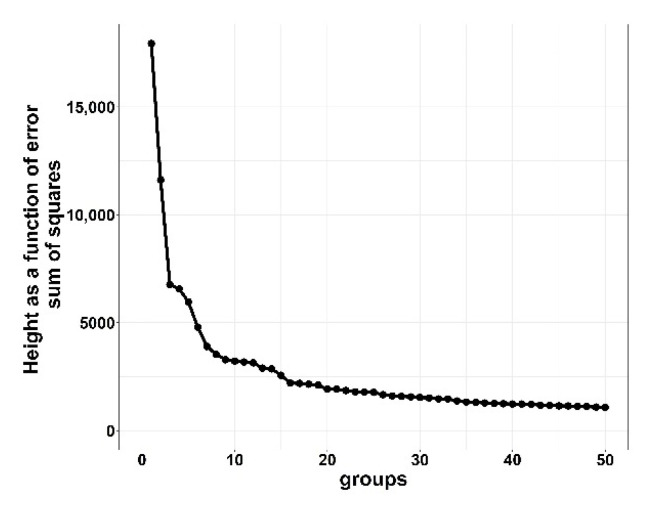
Elbow plot of LOW (>0 to <15%) bovine respiratory disease (BRD) morbidity cohorts demonstrating height as a function of error sum of squares for clustering cohorts with bovine respiratory disease from 10 US commercial feedlots. Distance between points is a difference in height of error sum of squares. Greater heights are suggestive of larger differences between groups.

**Figure 3 vetsci-10-00089-f003:**
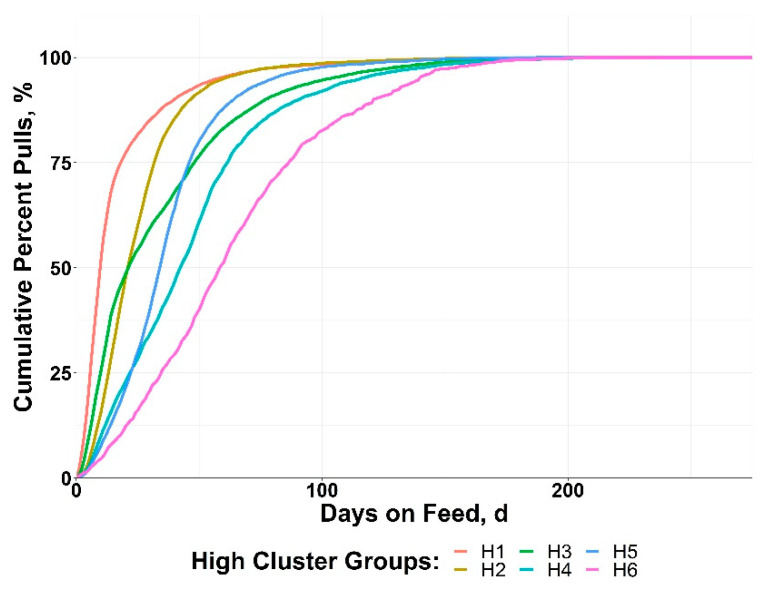
Data from 10 US feedlots were used to perform a cluster analysis of cohorts with ≥15% bovine respiratory disease (BRD) morbidity based on the cumulative timing, days on feed (DOF).

**Figure 4 vetsci-10-00089-f004:**
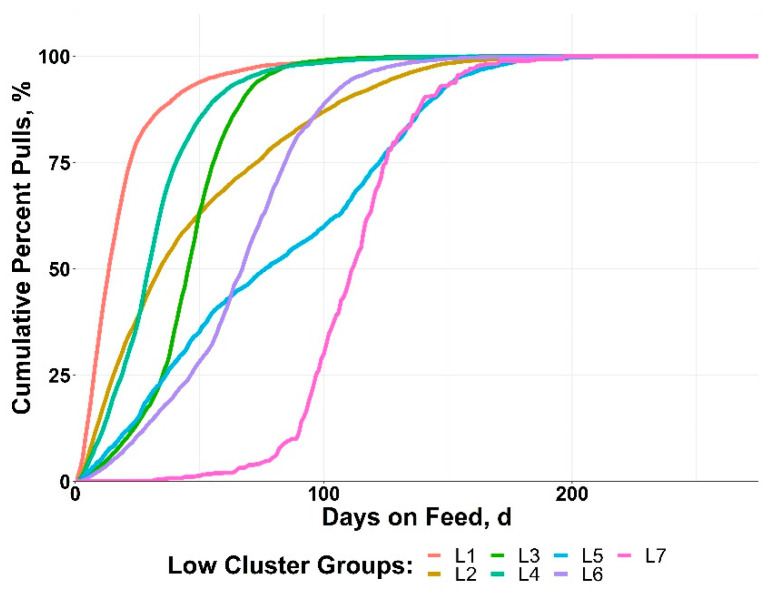
Data from 10 US feedlots were used to perform a cluster analysis of cohorts with >0 to <15% bovine respiratory disease (BRD) morbidity based on the cumulative timing, days on feed (DOF).

**Table 1 vetsci-10-00089-t001:** Descriptive statistics of HIGH, LOW, and ZERO cumulative bovine respiratory disease morbidity clusters and risk factors. Data from 10 US commercial feedlots were used to analyze hierarchical clustering of cumulative distributions of first treatment BRD in HIGH (≥15%) LOW (<15%) or ZERO (0%) cohorts.

	HIGH						LOW							ZERO
Cluster Class	H1	H2	H3	H4	H5	H6	L1	L2	L3	L4	L5	L6	L7	0
Lots (n)	352	437	246	225	267	70	1163	1428	734	1392	385	717	84	235
heifers	201	278	160	155	164	42	696	897	471	857	232	503	58	120
steers	151	159	86	70	103	28	467	531	263	535	153	214	26	115
Average Arrival Wt, kg	318	316	312	303	322	305	347	341	351	349	337	345	364	387
SD	49	47	42	45	41	48	51	44	43	44	45	47	38	42
Average Shrink, %	3.49	3.16	3.21	3	2.86	1.76	2.05	2	2	1.92	1.32	1.54	1.39	1.04
SD	2.72	2.33	2.32	2.57	2.18	2.21	2.46	2.43	2.48	3.64	2.81	2.47	2.38	2.28
Average number Cattle received	111	116	119	120	117	105	130	145	133	139	138	139	117	84
SD	49	54	48	47	52	46	77	65	63	72	59	64	56	34
Received Cattle														
<75 cattle	109	129	56	50	80	19	342	273	187	296	65	147	29	135
76 to 129	122	136	90	76	72	31	317	339	187	371	115	204	28	70
130 to 167	76	97	63	57	66	13	213	304	144	285	82	142	11	20
>167	45	75	37	42	49	7	291	512	216	440	123	224	16	10
Arrival Quarter														
Quarter 1	71	99	61	63	87	17	331	351	193	326	118	175	30	54
Quarter 2	37	90	54	60	78	24	228	433	244	394	123	238	24	60
Quarter 3	88	92	64	62	60	19	329	385	215	456	68	200	18	73
Quarter 4	156	156	67	40	42	10	275	259	82	216	76	104	12	48
Average Death Loss, %	4.17	4.21	2.93	2.84	3.06	2.09	1.24	1.44	1.28	1.3	1.38	1.26	1.23	0.51
SD	4.86	4.45	2.28	2.29	2.56	1.57	1.37	1.3	1.34	1.32	1.26	1.21	1.48	0.83
Average Morbidity	31.03	27.86	24.15	22.05	25.3	20.25	6.08	6.83	5.69	6.56	5.08	4.89	2.33	0
SD	13.25	11.79	8.39	6.76	9.97	4.79	4.18	3.49	3.76	3.95	2.98	3.3	1.75	0
Average BRD to 50%	9.76	21.57	21.25	40.8	34.38	61.33	13.22	31.22	44.37	27.65	64.55	61.79	110.76	0
Median BRD to 50%	10	21	22	41	34	57	13	31	44	28	61	61	109	0
SD	3.52	4.96	7.24	7.91	4.74	19.09	5.92	13.31	7.45	7.65	27.63	14.9	19	0
Mean Railers per cohort	1.78	2.42	1.62	1.56	1.68	0.77	0.64	0.82	0.66	0.78	0.57	0.48	0.25	0.18
Median Railers per cohort	1	1	1	1	1	0	0	0	0	0	0	0	0	0
SD	2.34	7.33	2.15	1.75	1.94	1.14	1.18	1.96	1.24	1.39	1.15	0.88	0.56	0.44
Mean days on feed, DOF	169	157	176	183	171	181	145	162	153	152	168	156	148	137
Median DOF	168	161	173	179	168	178	146	158	151	151	160	151	145	137
SD	37	38	31	33	31	29	32	27	27	26	31	28	25	18
Mean ADG	1.55	1.63	1.69	1.54	1.64	1.44	1.61	1.61	1.58	1.56	1.61	1.62	1.61	1.58
SD	0.84	1.50	1.66	0.65	1.52	0.26	1.85	1.06	0.67	0.50	1.10	0.86	0.49	0.20

**Table 2 vetsci-10-00089-t002:** Univariate model estimates of cohort demographics and their association by temporal clustering group using data from 10 US commercial feedlot operations. Feedlot cattle were grouped into HIGH (≥15%), LOW (>0 to <15%), or ZERO (0%) BRD cohort morbidity for clustering analysis. Differing letters within the same row indicate a statistical difference of *p* < 0.05 while estimates sharing a letter are not statistically different *p* > 0.05.

BRD Status	HIGH						LOW							ZERO
Cluster	H1	H2	H3	H4	H5	H6	L1	L2	L3	L4	L5	L6	L7	0
Cattle received ^1^	114 ^b^	119 ^b^	121 ^bcd^	123 ^bcde^	120 ^bc^	103 ^ab^	134 ^cde^	145 ^g^	135 ^defg^	143 ^fg^	137 ^cdefg^	137 ^efg^	117 ^abcde^	93 ^a^
SE	7.86	7.71	8.16	8.25	8.07	10.30	7.34	7.31	7.48	7.31	7.81	7.50	9.85	8.22
Shrink, % ^2^	3.65 ^f^	3.26 ^ef^	3.47 ^f^	3.08 ^ef^	2.96 ^ef^	2.13 ^abcde^	2.27 ^cd^	2.30 ^d^	2.24 ^bcd^	2.19 ^bcd^	1.68 ^ab^	1.90 ^abc^	1.74 ^abcd^	1.24 ^a^
SE	0.32	0.30	0.30	0.32	0.32	0.31	0.41	0.28	0.28	0.29	0.28	0.30	0.29	0.39
Arrival weight, kg	309 ^bc^	315 ^c^	302 ^ab^	295 ^a^	312 ^bc^	295 ^ab^	340 ^ef^	333 ^d^	341 ^fg^	341 ^f^	332 ^de^	338 ^def^	358 ^gh^	376 ^h^
SE	8.3	8.1	8.0	8.3	8.3	8.2	9.4	7.9	7.9	7.9	7.9	8.1	7.9	9.1
Prob of being Steer ^3^	0.40 ^b^	0.35 ^b^	0.31 ^ab^	0.28 ^ab^	0.35 ^ab^	0.36 ^ab^	0.35 ^b^	0.33 ^ab^	0.32 ^ab^	0.35 ^ab^	0.35 ^ab^	0.27 ^a^	0.27 ^ab^	0.41 ^b^
SE	0.04	0.04	0.04	0.04	0.04	0.06	0.03	0.03	0.03	0.03	0.04	0.03	0.05	0.04

^1^ Cattle Received = Model-estimated number of cattle received per clustering group. ^2^ Shrink = Model-estimated shrink (difference in pay weight from average cohort arrival weight × 100) expressed as percent. ^3^ Probability of being steer = Model-estimated probability of a cohort being steer within clustering group.

**Table 3 vetsci-10-00089-t003:** Model estimates of performance and health metrics and their association by temporal clustering group using data from 10 US commercial feedlot operations. Feedlot cattle were grouped into HIGH (≥15%), LOW (>0 to <15%), or ZERO (0%) BRD cohort morbidity for clustering analysis. Differing letters within the same row indicate a statistical difference of *p* < 0.05 while estimates sharing a letter are not statistically different *p* > 0.05.

BRD Status	High						Low							Zeros
Cluster	H1	H2	H3	H4	H5	H6	L1	L2	L3	L4	L5	L6	L7	0
ADG, kg ^1^	1.41 ^ab^	1.46 ^abc^	1.46 ^abcd^	1.42 ^ab^	1.45 ^abc^	1.33 ^a^	1.51 ^bc^	1.57 ^de^	1.58 ^e^	1.57 ^de^	1.55 ^cde^	1.59 ^e^	1.63 ^cde^	1.66 ^e^
SE	0.04	0.04	0.04	0.04	0.04	0.05	0.04	0.04	0.04	0.04	0.04	0.04	0.06	0.05
BRD Morb, % ^2^	27.25 ^j^	25.37 ^i^	21.07 ^gh^	19.43 ^fg^	22.53 ^h^	17.69 ^f^	5.73 ^d^	6.47 ^e^	5.42 ^d^	6.29 ^e^	4.68 ^c^	4.73 ^c^	2.33 ^b^	0.01 ^a^
SE	0.82	0.77	0.76	0.74	0.77	0.93	0.34	0.36	0.34	0.36	0.34	0.32	0.32	0.02
Death loss, % ^3^	4.19 ^e^	4.15 ^e^	2.94 ^cd^	2.76 ^cd^	3.06 ^d^	2.08 ^bc^	1.53 ^b^	1.65 ^b^	1.52 ^b^	1.54 ^b^	1.56 ^b^	1.54 ^b^	1.64 ^ab^	0.93 ^a^
SE	0.26	0.25	0.26	0.27	0.26	0.33	0.24	0.24	0.24	0.24	0.24	0.24	0.32	0.27
Rails/Culls ^4^	11.9 ^gh^	13.6 ^h^	11.2 ^fhg^	9.9 ^fg^	6.6 ^f^	7.6 ^efg^	4.0 ^b^	5.1^de^	4.2 ^bc^	4.4 ^bc^	5.3 ^cde^	3.6 ^b^	2.8 ^abcd^	1.8 ^a^
SE	3.3	3.7	2.7	2.7	2.6	2.3	1.1	1.4	1.1	1.2	1.5	1.0	1.0	0.5
Days to 50% BRD ^5^	9.4 ^b^	21.3 ^d^	20.5 ^d^	40.3 ^h^	34.2 ^g^	59.8 ^j^	12.5 ^c^	29.5 ^f^	44.0^i^	27.0 ^e^	61.2 ^j^	60.6 ^j^	109.8 ^k^	0.0 ^a^
SE	0.33	0.45	0.57	0.84	0.71	1.77	0.24	0.34	0.53	0.33	0.82	0.63	2.19	0

^1^ ADG = Model estimated Average Daily Gain (kg/day) = (Ship weight − Arrival weight)/days on fee.d. ^2^ BRD Morb = Model-estimated cohort bovine respiratory disease morbidity expressed as percentage of cattle received. ^3^ Death Loss = Model-estimated total death less per cohort expressed as percent of cattle received. ^4^ Rails/Culls = Model-estimated rails/culls (cattle removed from cohort prior to shipment for harvest) per 1000 cattle received. ^5^ Days to 50% BRD = Model-estimated days on feed for a cohort within cluster to reach 50% of their respective BRD first treatments.

## Data Availability

Data utilized for this research were from cooperating entities and are not available publicly due to confidentiality and anonymity agreements.

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
