# Peer review of "An Evaluation of Temporal Distributions of High, Low, and Zero Cohort Morbidity of Cumulative First Treatment Bovine Respiratory Disease and Their Associations with Demographic, Health, and Performance Outcomes in US Feedlot Cattle"

_vetsci, 2023, doi:10.3390/vetsci10020089_

Round 1

Reviewer 1 Report

The article An evaluation of temporal distributions of high, low, and zero cohort morbidity of cumulative first treatment bovine respiratory disease and their associations with demographic, health, and performance outcomes in US feedlot cattle, by B. Johnson et al. has been reviewed. The main objective of present study is to describe the BRD which occurs during the feeding period while incorporating the two factors, the magnitude of cohort morbidity and the timing of BRD occurred. Although their attempt is very interesting, I think that further discussions and refinement of the manuscript are necessary before this paper can be published.

By pre-dividing the cohort into three groups according to BRD morbidity, the authors were able to demonstrate differences in BRD-related factors among the groups (HIGH, LOW and ZERO) using descriptive statistics. I believe that this result provides an important contribution to this study area. However, I couldn’t understand significance of performing cluster analysis. As the authors cited, the previous study conducted by Babcock et al. revealed the association between timing of BRD and cattle health and performance by combination usage of cluster analysis and risk factor analysis (Babcock et al. 2010). Although almost same analysis has been conducted in this study, the authors did not fully discuss what the cluster analysis results showed. Since there are many previous studies which used methods similar to the present study such as Babcock et al (2010) and M. E. Theurer et al (2021), it would be very important to fully explain the originality or novelty of this study to the readers.

Moreover, I found several careless misses such as misspelling and inappropriate sentences due to copy-paste errors (“Error! Reference source not found.”) in this article. The order of the tables and their footnotes were also inappropriate. Boxplot Therefore, I strongly recommend the authors to read the instructions for authors carefully and to revise the entire manuscript appropriately with the help of someone experienced in writing academic papers. Besides, I think and boxplot with beeswarm plot would be suitable for showing the distribution of values in each cluster and significant differences of the values between clusters.

Author Response

Comments and Suggestions for Authors

The article “An evaluation of temporal distributions of high, low, and zero cohort morbidity of cumulative first treatment bovine respiratory disease and their associations with demographic, health, and performance outcomes in US feedlot cattle”, by B. Johnson et al. has been reviewed. The main objective of present study is to describe the BRD which occurs during the feeding period while incorporating the two factors, the magnitude of cohort morbidity and the timing of BRD occurred. Although their attempt is very interesting, I think that further discussions and refinement of the manuscript are necessary before this paper can be published.

AU: we appreciate the reviewers comments and have modified the manuscript to address the specific comments as cited below. Additionally we have refined the manuscript and added to the discussion as suggested.

By pre-dividing the cohort into three groups according to BRD morbidity, the authors were able to demonstrate differences in BRD-related factors among the groups (HIGH, LOW and ZERO) using descriptive statistics. I believe that this result provides an important contribution to this study area. However, I couldn’t understand significance of performing cluster analysis. As the authors cited, the previous study conducted by Babcock et al. revealed the association between timing of BRD and cattle health and performance by combination usage of cluster analysis and risk factor analysis (Babcock et al. 2010). Although almost same analysis has been conducted in this study, the authors did not fully discuss what the cluster analysis results showed. Since there are many previous studies which used methods similar to the present study such as Babcock et al (2010) and M. E. Theurer et al (2021), it would be very important to fully explain the originality or novelty of this study to the readers.

AU: we agree this work builds on previous studies related to disease timing. One of the areas we think this work is novel is the division by magnitude of morbidity (which has not been performed in previous work) allowing distinction between timing of morbidity and frequency of cases.the results group into morbidity levels and timing patterns which could be useful for future work evaluating potential methods to predict the disease pattern and magnitude within a group.  To further address this in the manuscript, we have added the following:  

Another novelty of the current research was incorporating cohorts reporting no BRD morbidity as well as not limiting the timing of BRD for inclusion of cohorts. This allowed for comparisons of cohorts to be grouped by the timing of BRD or lack thereof and determine if there were differences between cohort demographics and performance of cattle within the cohorts fitting each group based on the timing of magnitude of BRD.

Moreover, I found several careless misses such as misspelling and inappropriate sentences due to copy-paste errors (“Error! Reference source not found.”) in this article.

AU: we apologize for the formatting of the manuscript. We have corrected this error message as it was due to our reference manager.  

 The order of the tables and their footnotes were also inappropriate. Boxplot ?? (author) Therefore, I strongly recommend the authors to read the instructions for authors carefully and to revise the entire manuscript appropriately with the help of someone experienced in writing academic papers.

AU: We agree the tables and footnotes should have been placed near corresponding text and we have adjusted the order of tables and footnotes to match the instructions to authors. We have tried to revise the manuscript for both readability and clarity.

Besides, I think and boxplot with beeswarm plot would be suitable for showing the distribution of values in each cluster and significant differences of the values between clusters.

AU: we agree that boxplots would help with data representation, however we do not have room in the manuscript to illustrate boxplots on all relevant variables and elected to include a table. The table of descriptive statistics (TABLE 1 for each group of cohorts) which includes SE of the observations. Tables 2 and 3 show statistical comparisons on univariate and multivariate comparisons with SE. Ultimately, to this author, these tables describe the same wishes as requested by Reviewer 1 in a denser format for the purposes of not creating over 10 figures.

Reviewer 2 Report

This manuscript described the temporal patterns of BRD treatment in HIGH and LOW morbidity cohorts of feedlot calves through cluster analysis and their association with demographic, overall health, and performance outcomes. The information generated through this research is highly innovative and needed in the industry. 

Author Response

This manuscript described the temporal patterns of BRD treatment in HIGH and LOW morbidity cohorts of feedlot calves through cluster analysis and their association with demographic, overall health, and performance outcomes. The information generated through this research is highly innovative and needed in the industry. 

AU: we appreciate the comments and support for the manuscript.

Reviewer 3 Report

vetsci-2089442 An evaluation of temporal distributions of high, low, and zero cohort morbidity of cumulative first treatment bovine respiratory disease and their associations with demographic, health, and performance outcomes in US feedlot cattle

This article evaluated the association of cohorts relative to several aspects of bovine respiratory disease (BRD). The methodology used is acceptable but the major problem, in my opinion is the way in which the methodology and results are presented, which must be improved before I can recommend this paper for publication.

Major problems

L106, how were these groups determined (i.e., were specify indices used to determine e high and low morbidity).

L126-127, what criteria were used to arrive at these 45.5 kg intervals?

L128-129. Authors should provide a brief explanation as to the significance of these 13 cohorts.

Results: It is recommended that the authors standardize the number of decimals used in percentages (one or two, but not during the entire paper). Additionally, authors should present the percentage as well as the abstract values  (e.g., 50%; 200/400) so that readers can easily understand what is being presented.

Furthermore, the results section is somewhat confused and should be revised for clarity. There are several tables and figures in the submitted paper, but none of these were associated with the resulted presented. This in part resulted to the lack of clarity in the data presented, since the reader is obliged to try to decide which table is related to a particular result.

L167-168. how can you possibly present this data without first indicating the total number of cohorts per each group? How did you arrive at these indices? If there are present in any table or figure, then this must be provided here.

L269-270. This information (definition of high and low morbidity) should have been provided in the MM section since this only is known here at the discussion.

Minor problems

L305, it is not clear if you are referring to the cited authors (19 and 20) or the authors of the current study. Please clarify.

References #19 and 33 are the same. Please correct.

Author Response

vetsci-2089442 An evaluation of temporal distributions of high, low, and zero cohort morbidity of cumulative first treatment bovine respiratory disease and their associations with demographic, health, and performance outcomes in US feedlot cattle

 This article evaluated the association of cohorts relative to several aspects of bovine respiratory disease (BRD). The methodology used is acceptable but the major problem, in my opinion is the way in which the methodology and results are presented, which must be improved before I can recommend this paper for publication.

AU: we appreciate your review and feedback. We have tried to address the concerns through manuscript revisions and have provided specific corrections in more detail below.

Major problems

L106, how were these groups determined (i.e., were specify indices used to determine e high and low morbidity).

AU: The groups were determined using a 15% BRD morbidity threshold similar to reported in previous work (Rojas et al). The paragraph was re-written and there is further discussion of the 15% cutoff in the discussion section of the manuscript.

L126-127, what criteria were used to arrive at these 45.5 kg intervals?

AU: the groups were based on conversion from standard US measurements (pounds) and relevant 100 lb categories for the industry.  The categories represent weight classes of cattle which often receive different prices and would be externally valid for readers. Average cohort arrival weight was categorized into seven 45.5kg categories (“181kg to 226kg”, “227kg to 272kg”, “272.1kg to 317kg”, “318kg to 362kg”, “363kg to 408kg”, “408kg to 453kg”, “454kg to 498kg”) to avoid lack of model convergence and for external generalization. 45.5kg weight groups are commonly used to describe cohorts of cattle within the feeding industry.

L128-129. Authors should provide a brief explanation as to the significance of these 13 cohorts.

AU: we have added to the results section a  few paragraphs describing more of the temporal differences among the clusters and potential meaningfulness of these differences. Additionally, we have associated these paragraphs with the two figures (3 and 4) describing timing of clusters.

Results: It is recommended that the authors standardize the number of decimals used in percentages (one or two, but not during the entire paper). Additionally, authors should present the percentage as well as the abstract values  (e.g., 50%; 200/400) so that readers can easily understand what is being presented.

AU: thank you for catching this; we have standardized to two decimals in all of the tables for the percentages.

Furthermore, the results section is somewhat confused and should be revised for clarity. There are several tables and figures in the submitted paper, but none of these were associated with the resulted presented. This in part resulted to the lack of clarity in the data presented, since the reader is obliged to try to decide which table is related to a particular result.

AU: we agree and the results section has been revised placing tables / figures near the text where relevant and referenced each of them appropriately.  We have also clarified the results section throughout. You are correct that figures 3 and 4 were not addressed in the results due to inadvertent deletion of a paragraph: this has now been rectified and all tables and figures are addressed in the text.

L167-168. how can you possibly present this data without first indicating the total number of cohorts per each group? How did you arrive at these indices? If there are present in any table or figure, then this must be provided here.

AU: we have provided the total number of cohorts utilized for the analysis: Line 163 provides the total number of cohorts in the analysis (7,735). Line 163 states 5,903 cohorts were in LOW, 5,903 / 7,735 = 76.3%. There were 1,597 cohorts in the high (1,597/7,735 = 20.6%) and finally 235 cohorts in the ZERO category (235/7,735 = 3.01%). The paragraph was re-structured for clarification. These numbers are presented early int eh results section and in table 1 the specific number of cohorts per cluster is defined.

L269-270. This information (definition of high and low morbidity) should have been provided in the MM section since this only is known here at the discussion.

 AU: Cohorts were defined as high, low, or zero in MM lines 97-105. This paragraph has been restructured for clarity.

Minor problems

L305, it is not clear if you are referring to the cited authors (19 and 20) or the authors of the current study. Please clarify.

AU: we have attempted to clarify this statement:

Findings showing higher levels of shrink associated with higher morbidity was similar to previous research, which reported that cattle with greater shrink (body weight loss from purchase to feedlot arrival) had greater BRD morbidity risk as well as lower arrival body weights[19,20].

References #19 and 33 are the same. Please correct.

AU: thank you for identifying this; we have corrected (removed 33)

Round 2

Reviewer 1 Report

The authors corrected their previous manuscript appropriately and added new discussions. So I think the revised manscript could be accepted.

Reviewer 3 Report

The authors have responded satisfactorily to all my doubts, and I have no further issue relative to the publication of this paper.